# Dearomative 1,4-difunctionalization of naphthalenes via palladium-catalyzed tandem Heck/Suzuki coupling reaction

Bo Zhou[1,5], Hongliang Wang[2,5], Zhong-Yan Cao[1], Jia-Wen Zhu[1], Ren-Xiao Liang[1], Xin Hong [2,3✉] & Yi-Xia Jia [1,4✉]

Dearomative functionalization reactions represent an important strategy for the synthesis of valuable three-dimensional molecules from simple planar aromatics. Naphthalene is a challenging arene towards transition-metal-catalyzed dearomative difunctionalization reactions. Reported herein is an application of naphthalene as a masked conjugated diene in a palladium-catalyzed dearomative 1,4-diarylation or 1,4-vinylarylation reaction via tandem Heck/Suzuki sequence. Three types of 1,4-dihydronaphthalene-based spirocyclic compounds are achieved in excellent regio- and diastereoselectivities. Key to this transformation is the inhibition of a few competitive side reactions, including intramolecular naphthalenyl C-H arylation, intermolecular Suzuki cross-coupling, dearomative 1,2-difunctionalization, and dearomative reductive-Heck reaction. Density functional theory (DFT) calculations imply that the facile exergonic dearomative insertion of a naphthalene double bond disrupts the sequence of direct Suzuki coupling, leading to the tandem Heck/Suzuki coupling reaction. The observed regioselectivity towards 1,4-difunctionalization is due to the steric repulsions between the introduced aryl group and the spiro-scaffold in 1,2-difunctionalization.

[1] College of Chemical Engineering, State Key Laboratory Breeding Base of Green-Chemical Synthesis Technology, Zhejiang University of Technology, Hangzhou 310014, China. [2] Department of Chemistry, Zhejiang University, Hangzhou 310058, China. [3] State Key Laboratory of Clean Energy Utilization, Zhejiang University, Zheda Road 38, Hangzhou 310027, China. [4] State Key Laboratory of Organometallic Chemistry, Shanghai Institute of Organic Chemistry, Chinese Academy of Sciences, Shanghai 200032, China. [5] These authors contributed equally: Bo Zhou, Hongliang Wang. ✉email: hxchem@zju.edu.cn; yxjia@zjut.edu.cn

Owing to the facile assembly of complex molecules starting from readily available olefins, transition-metal-catalyzed olefin difunctionalization has received considerable attention in past decades[1–9]. In this context, domino transformations involving initial carbometalation of simple alkenes or conjugated dienes and subsequent capture of the in situ generated σ-alkylmetal or π-allyl-metal species has been intensely developed for this purpose (Fig. 1a)[10–16]. On the other hand, as disclosed recently by the groups of Lautens[17–20], Liang[21], Yin[22,23], Zhou[24], Wu[25], and us[26–30], the application of endocyclic C=C bonds of heteroarenes as non-classic olefins has enabled a number of efficient dearomative difunctionalization reactions of indoles and furans with Pd- or Ni-catalyst[31]. This dearomatizing strategy undoubtedly expands the scope of olefin difunctionalization reaction to arenes and constitutes an important method for the synthesis of valuable three-dimensional molecules from simple planar aromatics[32–38]. However, the present study is still in its infancy and is limited to reactive heteroarenes, which tend to dearomatization due to their generally lower resonance stabilization energy. It is highly desirable to extend this reaction to additional aromatic compounds, in particular, for those less reactive arenes.

Naphthalene and benzene are abundant aromatic molecules while remain much less reactive toward the above-mentioned dearomative difunctionalization reactions[34–39]. When viewing naphthalene as a masked conjugated diene, not only dearomatizing carbometalation but also regioselective and diastereoselective capture of the in situ formed π-allyl-metal species remain challenging (Fig. 1b). In addition, competitive side reactions

including naphthalenyl C–H functionalization and the direct cross-coupling with capturing agents impeded this study. To our knowledge, there has no example reported for the dearomative difunctionalization of naphthalenes through palladium-catalyzed Heck/anionic-capture sequence[40–50]. As our interest of dearomatization reactions[26–30,51–54], we report herein a dearomative 1,4-difunctionalization of naphthalenes via intramolecular dearomative Heck arylation or vinylation followed by intermolecular cross-coupling with the π-allyl-palladium species (Fig. 1c). A range of 1,4-dihydronaphthalene-based spirocycles bearing oxindole, dihydrobenzofuran, or indene subunits are achieved in moderate to excellent yields, which are frequently occurring structural frameworks in natural products and bioactive molecules. Both regioselectivity and remote diastereoselectivity of two newly formed carbocenters at C1 and C4 are well controlled. Key to this reaction relies on the inhibition of a few competitive side reactions, including intramolecular naphthalenyl C–H arylation, intermolecular Suzuki cross-coupling, 1,2-difunctionalization, and dearomative reductive-Heck reaction. DFT calculations reveal the mechanistic basis for the tandem Heck/Suzuki coupling reaction and the controlling factors of chemo- and regioselectivities.

## Results

**Optimization study**. Our study began with the reaction optimization of N-(2-bromophenyl)-N-methyl-1-naphthamide **1a** with phenylboronic acid **2a** (Table 1). It has turned out that 1,4-product **3a** was difficult to be achieved as a series of by-products **4–7** were observed in the reaction. As shown in Fig. 2, initial test in

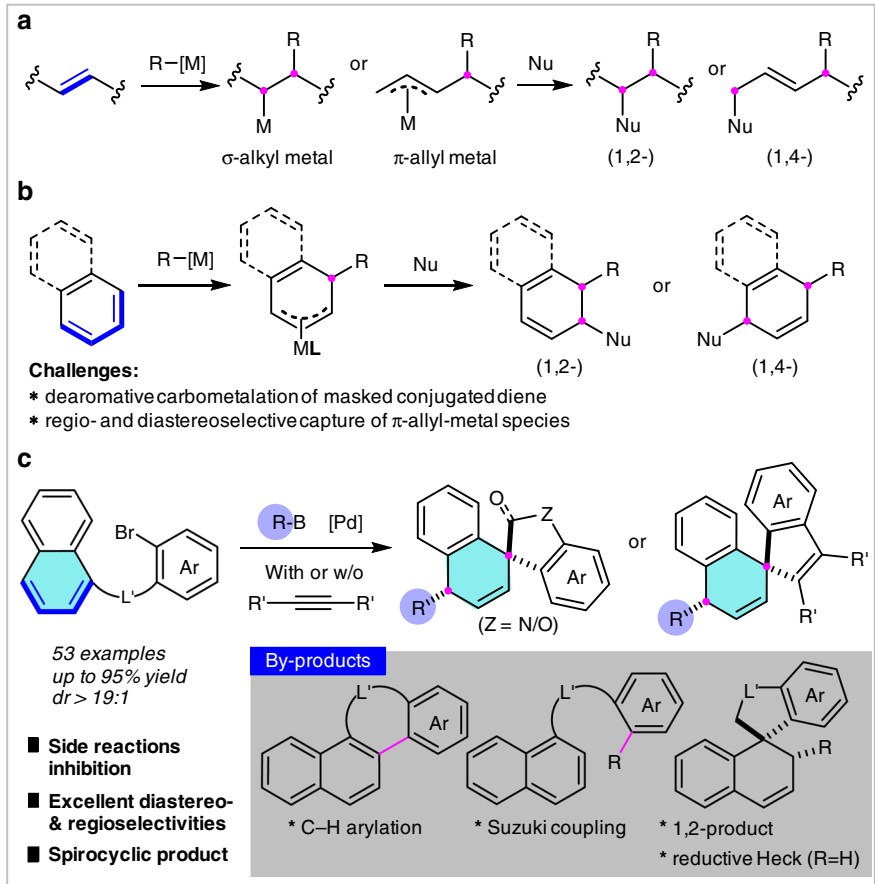

**Fig. 1 Transition-metal-catalyzed difunctionalization of olefin and arene. a** TM-catalyzed difunctionalization of alkene (well-developed)[2,9–16]. **b** Dearomative difunctionalization of naphthalene or benzene (unknown). **c** Dearomative 1,4-diarylation/vinylarylation of naphthalenes via Heck/Suzuki coupling (This work). **[M]** metal catalyst, **Nu** nucleophile, **L** ligand, **L′** linker, **Ar** arene.

**Table 1 Reaction optimization[a].**

| Entry | Ligand | Base | Solvent | Yield of 3a (%)[b] | Yield of 4 (%)[b] | Yield of 5 (%)[b] | Yield of 6 (%)[b] | Yield of 7 (%)[b] |
|---|---|---|---|---|---|---|---|---|
| 1 | Ph$_3$P | K$_2$CO$_3$ | Toluene | 2 | 90 | 4 | – | – |
| 2 | Cy$_3$P | K$_2$CO$_3$ | Toluene | 7 | 91 | – | – | – |
| 3 | (EtO)$_3$P | K$_2$CO$_3$ | Toluene | 5 | 89 | – | – | – |
| 4 | **L1** | K$_2$CO$_3$ | Toluene | 16 | 61 | 7 | – | – |
| 5 | **L2** | K$_2$CO$_3$ | Toluene | 18 | 76 | – | – | – |
| 6 | **L3** | K$_2$CO$_3$ | Toluene | 24 | 54 | 6 | – | – |
| 7 | **L4** | K$_2$CO$_3$ | Toluene | 46 | 40 | 3 | 3 | – |
| 8 | **L5** | K$_2$CO$_3$ | Toluene | 42 | 52 | – | – | – |
| 9[c] | **L5** | Na$_2$CO$_3$ | Toluene | 18 | – | – | – | – |
| 10 | **L5** | Cs$_2$CO$_3$ | Toluene | – | 43 | 52 | – | – |
| 11 | **L5** | K$_3$PO$_4$ | Toluene | 21 | 66 | – | 10 | – |
| 12 | **L5** | KF | Toluene | 83 | 8 | 2 | 2 | – |
| 13 | **L5** | CsF | Toluene | 43 | 48 | 4 | 1 | 1 |
| 14 | **L4** | KF | Toluene | 71 | 21 | – | 3 | – |
| 15 | **L5** | KF | THF | 75 | 7 | 9 | 1 | – |
| 16 | **L5** | KF | DME | 72 | 9 | 11 | 2 | – |
| 17 | **L5** | KF | MeCN | 31 | 9 | 29 | 2 | 13 |
| 18 | **L5** | KF | DMF | 79 | 8 | 5 | – | – |
| 19[d] | **L5** | KF | Toluene | 46 | 45 | 6 | 1 | – |
| 20[e] | **L5** | KF | Toluene | 86 (79) | 8 | – | 2 | – |
| **21**[e,f] | **L5** | **KF** | **Toluene** | **91** (83) | **4** | **–** | **3** | **–** |
| 22[e,f] | **L5** | KF | Toluene | 89 | 3 | – | 4 | – |
| 23[e,f] | **L5** | KF | Toluene | 27 | – | – | – | – |

*THF* tetrahydrofuran, *DME* 1,2-dimethoxyethane, *DMF* N,N-dimethylformamide.
[a]Reaction conditions: Amide **1** (0.2 mmol), **2a** (0.4 mmol), Pd(dba)$_2$ (0.01 mmol), ligand (0.02 mmol), base (0.5 mmol), solvent (2.0 mL), at 120 °C, 24 h. Full conversion of **1a** unless otherwise noted (X = Br for entries 1–21; X = I for entry 22; X = Cl for entry 23).
[b]Determined by $^1$H NMR spectroscopy using MeNO$_2$ as an internal standard; isolated yield in parenthesis.
[c]23% conversion.
[d]At 100 °C.
[e]At 140 °C.
[f]0.20 mmol of Ph$_4$BNa and 16 h.

the presence of 5 mol% Pd(dba)$_2$, 10 mol% PPh$_3$, and 2.5 equiv K$_2$CO$_3$ in toluene led to by-product **4** in 90% NMR (Nuclear Magnetic Resonance) yield via an intermolecular Suzuki cross-coupling (see also entry 1). Not surprisingly, the intramolecular naphthalenyl C–H arylation reaction proceeded smoothly to afford compound **7** in 70% yield in the absence of phenylboronic acid **2a**. Although the desired product **3a** and by-product **5** were detected in very poor yields, it indicated the occurrence of naphthalene dearomatization.

Further ligand evaluation showed that the racemic BINOL ([1,1′-binaphthalene]-2,2′-diol)-derived phosphoramidite **L1** provided **3a** in a promising yield (16%) along with by-product **5** (7%) derived from dearomative reductive-Heck reaction. Cy$_3$P and (EtO)$_3$P gave similar results as PPh$_3$ (entries 2–4). Nevertheless, excellent diastereoselectivity (>19:1) was observed for **3a**, whose structure and the relative configuration were determined by X-ray analysis. Encouraged by this result, other racemic phosphoramidites **L2–L4** bearing different substituents on nitrogen atom were

**Fig. 2 Initial test with Ph₃P as ligand.** The yields refer to NMR yield using MeNO₂ as an internal standard. dba (1E,4E)-1,5-diphenylpenta-1,4-dien-3-one.

examined subsequently (entries 5–7). It was found that steric hindrance of the amino moiety facilitated the formation of **3a**. For example, **L4** bearing a dicyclohexylamino group led to **3a** in 46% yield along with the detection of by-products **4**–**6** (entry 7). Comparable yield of **3a** was achieved for **L5** containing a spirobackbone[55], while compound **4** was the only by-product in this case (entry 8).

To further improve the product yield, influence of the base and solvent were then investigated in the presence of ligand **L5** (entries 9–13 and 15–18). Gratifyingly, NMR yield of **3a** was substantially improved to 83% by employing KF as a base (entry 12), whereas no product or lower yields were observed for Na₂CO₃, Cs₂CO₃, K₃PO₄, and CsF (entries 9–11 and 13). As a comparison, the reaction using **L4** as a ligand in the presence of KF base led to **3a** in 71% yield (entry 14). Toluene was proved to be the best choice of solvent, although comparable yields were observed in THF, DME, and DMF (entries 15, 16, and 18). Noticeably, the reaction in MeCN led to relatively higher yields for by-products **5** and **7** (entry 17). Elevating the reaction temperature to 140 °C resulted in a slightly improved NMR yield of **3a** (86%), which was isolated in 79% yield (entry 20). Switching PhB(OH)₂ to Ph₄BNa further improved the isolated yield to 83%, along with trace amount of by-product **4** and 1,2-isomer **6** (entry 21). It is noting that lower enantioselectivities of the product were observed when the corresponding chiral ligands were used (entry 4 (S)-**L1**, 16% ee; entry 5 (R)-**L2**, 8% ee; entry 6 (S)-**L3**, 12% ee; entry 7 (S)-**L4**, 11% ee; entry 20 (R)-**L5**, 5% ee). As shown in entries 22 and 23, the influence of halides in the substrate **1** was investigated. Product **3a** was obtained in 89% NMR yield for iodo-substrate and 27% for the chloro-substrate.

**Substrate scope evaluation.** With the optimal conditions in hand, the scope of amide **1** was then evaluated (Fig. 3). Good yields and excellent diastereoselectivity (>19:1) were detected for the substrates bearing an aniline ring with different steric and electronic profiles. A variety of substituents, including Me, MeO, CF₃, F, Cl, CF₃O, acetyl, ester, and amide were well tolerated in the reaction. No steric effect was observed in the reactions of substrates bearing substituents *ortho* to bromide atom and the corresponding products **3b**–**3e** were achieved in good to excellent yields. In addition to *N*-methyl substrates, the reaction of *N*-benzyl substrate delivered product **3u** in a higher yield (95%). The reaction was also compatible with a pyridine-derived substrate, which afforded product **3v** in 72% yield. Moreover, for the reaction of substrate bearing an extra bromide atom on the naphthalene ring, an unsurprisingly diphenylated product **3x** was isolated in 71% yield with excess amount of Ph₄BNa and KF. Excellent yield was also observed for product **3w** containing a

methoxy group on naphthalene ring. Interestingly, although Cl atom at C5 of product **3j** survived in the reaction, Cl para to bromide coupled with Ph₄BNa to afford diphenylated product **3y** in 58% yield. In these two cases three C–C σ-bonds were generated in one step. Extension of the reaction to 1,4-difunctionalization of benzene was unsuccessful, while excellent yield of 88% was achieved for an antracene-substrate **1z**. When a *para*-MeO-substrate **8** was used, the 1,4-diarylation product **9** was isolated in 51% yield as a major product along with 1,2-product **10** in 22% yield.

Considering the commercial availability of aromatic boronic acids, we then applied them as coupling partners for this 1,4-difunctionalization reaction. As shown in Table 2, reactions of **1a** with various of boronic acids **2** bearing different substituents, such as Me, F, Cl, CF₃, and Ph, at *o-*/*m-*/*p*-positions of benzene ring proceeded well to afford the desired products in 50–90% yields with excellent diastereoselectivity (>19:1). Noteworthy is that a vinyl moiety bearing on the benzene ring of **2** has no disturbance to the reaction, affording product **3ai** in 46% yield (entry 9). Other than 1-naphthamide substrate, the reaction is also compatible with arylether tethered naphthalene **11**. Under the standard conditions, the anticipated 1,4-diarylation product **12** was afforded in 55% yield and with >19:1 diastereomeric ratio (Fig. 4). This not only extends the scope of naphthalene, but also provides a reliable synthetic approach to dihydrobenzofuran-derived spirocycle.

**Theoretical calculation study.** To shed light on the reaction mechanism and origins of chemo- and regioselectivities, density functional theory (DFT) calculations were performed using the *N*-(2-bromophenyl)-*N*-methyl-1-naphthamide **1a** and phenylboronic acid **2a** as the model. The computed free energy changes of the tandem Heck/Suzuki reaction (black pathway) and the intermolecular Suzuki cross-coupling reaction (blue pathway) are shown in Fig. 5. From the PdL₂ complex, ligand exchange with **1a** leads to the arene-coordinated intermediate **int1**. Subsequent oxidative addition via **TS2** generates the Pd(II) intermediate **int3**, which undergoes insertion through **TS4** to form the allyl-palladium intermediate **int5**. Transmetalation of **int5** with PhB(OH)₂F anion (see Supplementary Fig. 185 for the formation of PhB(OH)₂F anion) via **TS7** irreversibly affords the (allyl)palladium(phenyl) intermediate **int8**. Subsequent reductive elimination through **TS9** eventually produces the Heck/Suzuki product **3a** and regenerates the Pd(0) active catalyst. Alternatively, **int3** can also directly undergoes transmetalation with PhB(OH)₂F anion to form the biarylpalladium(II) intermediate **int12**. Subsequent aryl–aryl reductive elimination through **TS13** would produce the classic intermolecular Suzuki cross-coupling

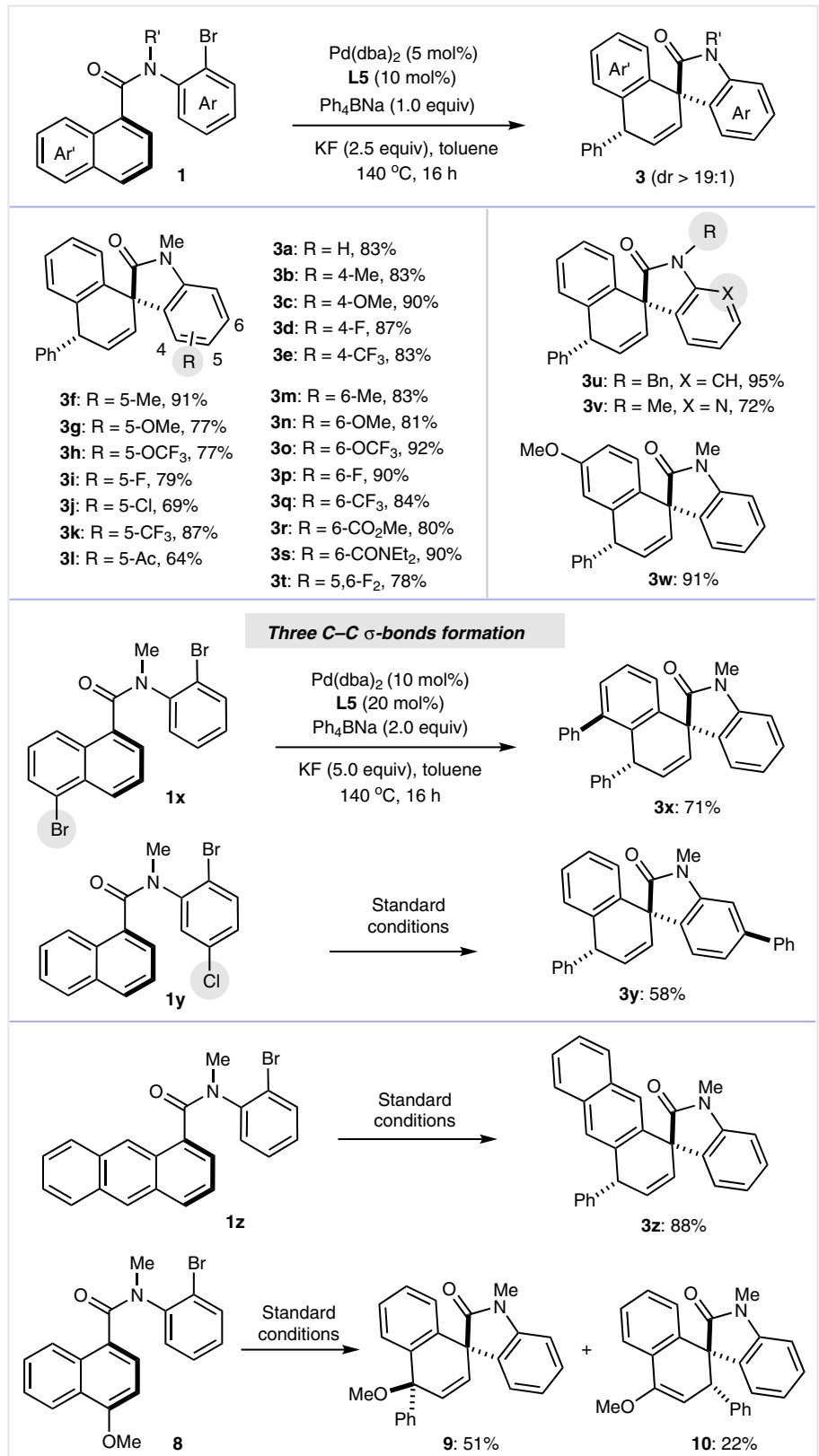

**Fig. 3 Substrate scope of amides.** The standard reaction conditions were used, as shown in Table 1, entry 21. Isolated yield are presented.

product **4**. The overall barrier for the formation of Heck/Suzuki product **3a** is 1.5 kcal/mol lower than that for the formation of **4** (**TS7** vs **TS11**), indicating that the formation of **3a** is more favorable than that of **4**. This is consistent with the experimental results that the ratio of **3a** and **4** is 86:8 (Table 1, entry 20). Therefore, the chemoselectivity-determining transmetalation step differentiates the tandem Heck/Suzuki coupling reaction and the competing classic intermolecular Suzuki cross-coupling. The

**Table 2 Scope of boronic acids[a].**

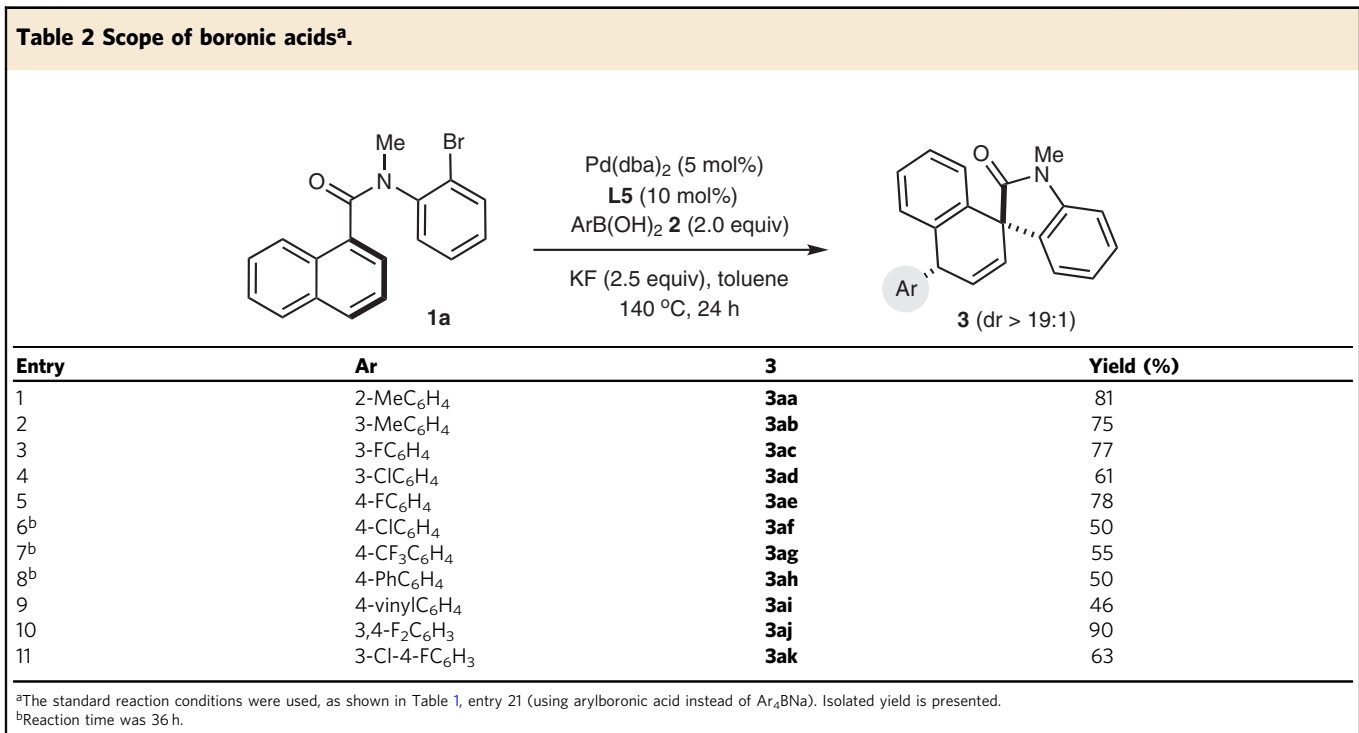

| Entry | Ar | 3 | Yield (%) |
|---|---|---|---|
| 1 | 2-MeC$_6$H$_4$ | 3aa | 81 |
| 2 | 3-MeC$_6$H$_4$ | 3ab | 75 |
| 3 | 3-FC$_6$H$_4$ | 3ac | 77 |
| 4 | 3-ClC$_6$H$_4$ | 3ad | 61 |
| 5 | 4-FC$_6$H$_4$ | 3ae | 78 |
| 6[b] | 4-ClC$_6$H$_4$ | 3af | 50 |
| 7[b] | 4-CF$_3$C$_6$H$_4$ | 3ag | 55 |
| 8[b] | 4-PhC$_6$H$_4$ | 3ah | 50 |
| 9 | 4-vinylC$_6$H$_4$ | 3ai | 46 |
| 10 | 3,4-F$_2$C$_6$H$_3$ | 3aj | 90 |
| 11 | 3-Cl-4-FC$_6$H$_3$ | 3ak | 63 |

[a]The standard reaction conditions were used, as shown in Table 1, entry 21 (using arylboronic acid instead of Ar$_4$BNa). Isolated yield is presented.
[b]Reaction time was 36 h.

**Fig. 4 1,4-Diarylation of arylether tethered naphthalene 11.** Reaction conditions: **11** (0.2 mmol), Pd(dba)$_2$ (0.01 mmol), **L5** (0.02 mmol), Ph$_4$BNa (0.2 mmol), KF (0.5 mmol), toluene (2 mL), 140 °C, 16 h, isolated yield.

facile exergonic insertion of pendant naphthalene double bond via **TS4** interrupts the sequence of Suzuki cross-coupling and drives the reaction toward the tandem Heck/Suzuki coupling reaction. We have also considered a number of alternative mechanistic pathways, and the details of these unfavorable pathways are included in the Supporting Information (Supplementary Figs. 187 and 188).

Based on the mechanistic model, we next explored the regioselectivity of 1,4- and 1,2-diarylation. The energies and optimized structures of the competing reductive elimination transition states are shown in Fig. 6 (see Supplementary Fig. 188 for the entire DFT-computed Gibbs free energy profile of the regioisomeric 1,2-difunctionalization). **TS9** is 3.0 kcal/mol more favorable than **TS14**, which is consistent with the experimental observations of 1,4-diarylation as the major product. The 1,2-diarylation is less favorable due to the steric repulsions between the phenyl moiety of the forming C–C bond and the spiro-scaffold. Indeed, replacing the spiro-scaffold with methylene led to a reversed chemoselectivity (see Supplementary Fig. 189). This suggested that the intrinsic chemoselectivity favors the 1,2-difunctionalization because of the conjugation in the styrene product, while the steric repulsions overrule the intrinsic preference and favors the 1,4-difunctionalization.

**Three-component dearomative reaction**. We further envisioned a three-component tandem reaction between 1-(2-bromophenyl)naphthalene **13**, alkyne **14**, and sodium tetraarylborates to synthesize indene-derived spirocarbocycles. In this dearomative 1,4-vinylarylation reaction, three C–C bonds would generate in one step through the first insertion of Ar–Pd species to alkyne, the second insertion of the resulting alkenyl–Pd intermediate to naphthalene, and the final transmetalation with tetraphenylborate followed by reductive elimination. To our delight, under the above optimal conditions, both aryl and alkyl substituted internal alkynes reacted smoothly with naphthalene **13** and Ph$_4$BNa, offering the spirocarbocycles **15a–15i** in moderate to good yields with excellent diastereoselectivities (Table 3, entries 1–9). Effect of the substituents attached on the benzene ring of arylbromides was also tested, which afforded the desired products **15j–15n** in the yields ranging from 55% to 83% (entries 10–13).

**Scale-up reaction and product transformation**. To demonstrate the practicality of this protocol, a scale-up reaction was carried out and product **3a** was isolated in 87% yield (Fig. 7a). As shown in Fig. 7b, non-conjugated carbon–carbon double bond of **3a** was prone to isomerization in the presence of $^t$BuOK, affording spirooxindole **16** bearing a styrene moiety in 91% yield.

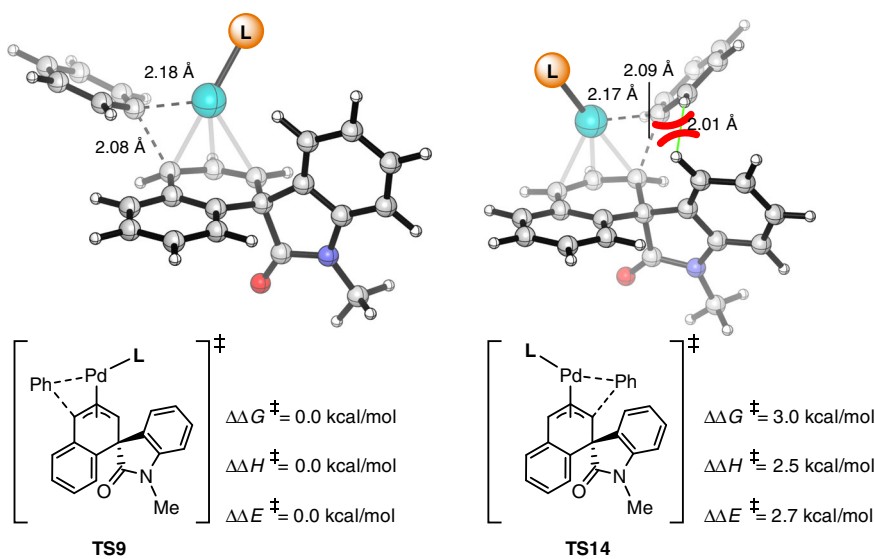

**Fig. 5 Theoretical calculations.** DFT-computed free energy profiles of the most favorable pathway for the formation of **3a** and by-product **4** (**L** = **L5**).

**Fig. 6 Theoretical calculations.** DFT-computed free energies of the regioselectivity-determining reductive elimination transition states (**L** = **L5**).

Moreover, a Pd/C-catalyzed hydrogenation of alkene with $H_2$ balloon led to saturated spirocarbocycle **17** in 92% yield. It is noted that compounds **3a**, **16**, and **17** are analogs of a patented bioactive molecule for treating pain[56], which indicates further potential application of the present method.

In summary, an efficient dearomative 1,4-diarylation of naphthalenes has been developed through palladium-catalyzed tandem Heck/Suzuki coupling reaction. A few competitive side reactions are overcome with excellent control of chemo-, regio-, and diastereoselectivities in this reaction. This protocol is further

**Table 3 Three-component dearomative 1,4-vinylarylation reaction[a].**

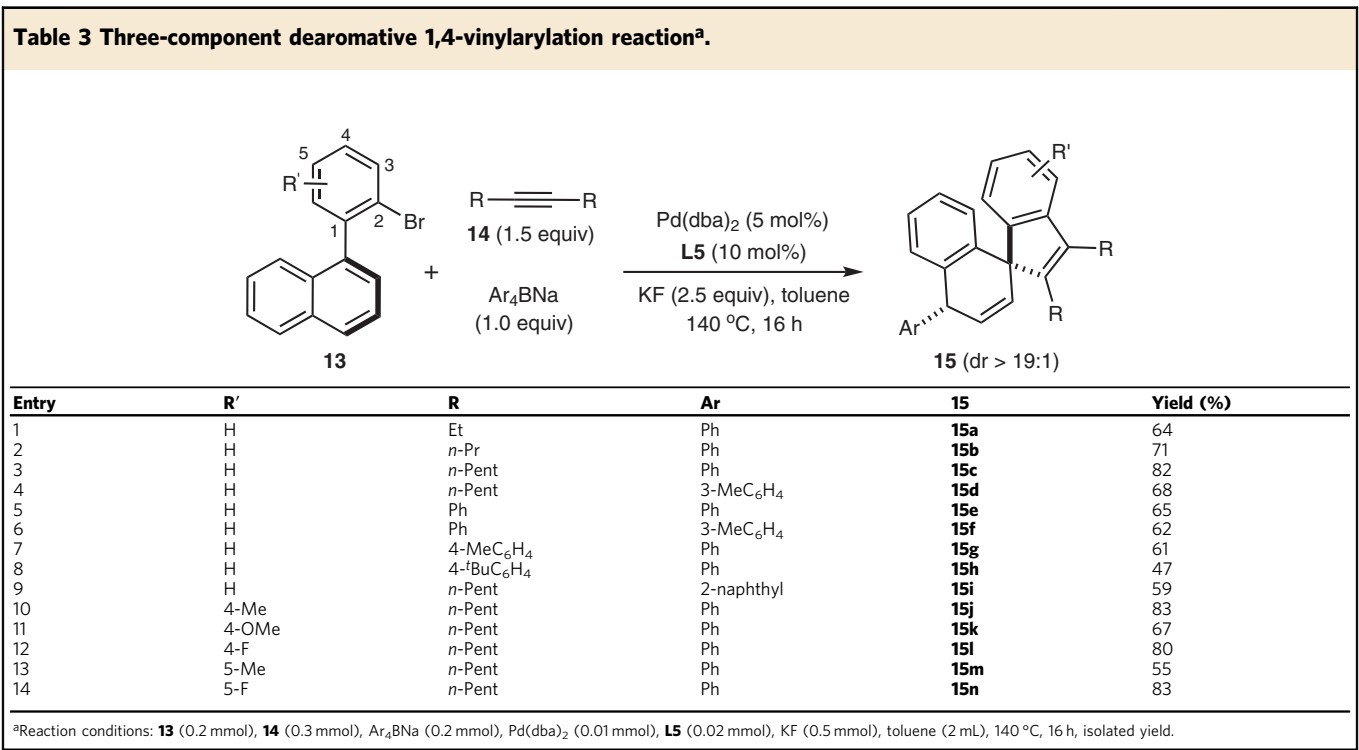

| Entry | R′ | R | Ar | 15 | Yield (%) |
|---|---|---|---|---|---|
| 1 | H | Et | Ph | 15a | 64 |
| 2 | H | n-Pr | Ph | 15b | 71 |
| 3 | H | n-Pent | Ph | 15c | 82 |
| 4 | H | n-Pent | 3-MeC$_6$H$_4$ | 15d | 68 |
| 5 | H | Ph | Ph | 15e | 65 |
| 6 | H | Ph | 3-MeC$_6$H$_4$ | 15f | 62 |
| 7 | H | 4-MeC$_6$H$_4$ | Ph | 15g | 61 |
| 8 | H | 4-$^t$BuC$_6$H$_4$ | Ph | 15h | 47 |
| 9 | H | n-Pent | 2-naphthyl | 15i | 59 |
| 10 | 4-Me | n-Pent | Ph | 15j | 83 |
| 11 | 4-OMe | n-Pent | Ph | 15k | 67 |
| 12 | 4-F | n-Pent | Ph | 15l | 80 |
| 13 | 5-Me | n-Pent | Ph | 15m | 55 |
| 14 | 5-F | n-Pent | Ph | 15n | 83 |

[a]Reaction conditions: **13** (0.2 mmol), **14** (0.3 mmol), Ar$_4$BNa (0.2 mmol), Pd(dba)$_2$ (0.01 mmol), **L5** (0.02 mmol), KF (0.5 mmol), toluene (2 mL), 140 °C, 16 h, isolated yield.

**Fig. 7 Scale-up reaction and synthetic transformations. a** 1.0 mmol scale-synthesis of product **3a**. **b** Transformations of product **3a**. (i) isomerization of carbon−carbon double bond of **3a** with $^t$BuOK. (ii) reduction of carbon−carbon double bond of **3a** with Pd/C catalyst under H$_2$ atmosphere.

extended to three-component dearomative 1,4-vinylarylation of naphthalenes with alkynes and tetraarylborates. The dearomative 1,4-difunctionalization reaction has provided facile accesses to unique spirocyclic compounds. DFT calculations revealed the reaction mechanism and elucidated the origins of chemo- and regioselectivities. It is the facile exergonic insertion of naphthalene double bond that disrupts the sequence of direct Suzuki coupling, leading to the tandem Heck/Suzuki coupling reaction. The steric repulsions between the aryl group of the forming C−C bond and the spiro-scaffold disfavors the intrinsic regioselectivity toward 1,2-difunctionalization, allowing the observed 1,4-difunctionalization.

## Methods

**General procedure for the reaction of 1, 8, or 11.** To a dried Schlenk tube were added **1**, **8**, or **11** (0.20 mmol), Pd(dba)$_2$ (5.8 mg, 0.010 mmol), ligand **L5** (9.2 mg, 0.020 mmol), NaBPh$_4$ (68.4 mg, 0.20 mmol) or ArB(OH)$_2$ (0.40 mmol), KF (29.1 mg, 0.50 mmol) under N$_2$. 2.0 mL toluene was then introduced via syringe and the

tube was sealed using Teflon cap. The mixture was stirred at 140 °C until the starting material was consumed. The solvent was then removed under vacuum and the residue was purified by chromatography on silica gel, eluting with ethyl acetate/petroleum ether to afford the products **3**, **9**, or **12**.

**General procedure for the reaction of 13**. To a dried Schlenk tube were added **13** (0.20 mmol) and **14** (0.30 mmol), Pd(dba)$_2$ (5.8 mg, 0.010 mmol), ligand **L5** (9.2 mg, 0.02 mmol), KF (29.1 mg, 0.50 mmol), Ar$_4$BNa (0.20 mmol) under N$_2$. 2.0 mL toluene was then introduced via syringe and the tube was sealed using Teflon cap. The mixture was stirred at 140 °C for 16 h. The solvent was then removed under vacuum and the residue was purified by chromatography on silica gel, eluting with dichloromethane/petroleum ether to afford the products **15**.

## Data availability

The X-ray crystallographic coordinate for structure reported in this study have been deposited at the Cambridge Crystallographic Data Centre (CCDC) under deposition numbers CCDC-1969160 (**3a**). These data can be obtained free of charge from The Cambridge Crystallographic Data Centre via www.ccdc.cam.ac.uk/data_request/cif. The authors declare that all other data supporting the findings of this study are available within the article and Supplementary information files, and also are available from the corresponding author upon reasonable request.

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

## Acknowledgements
We are grateful for the financial support from the National Natural Science Foundation of China (21772175 and 91956117, Y.-X.J.; 21702184, R.-X.L.; 21702182 and 21873081, X.H.), China Postdoctoral Science Foundation (2019M652056, H.W.), Fundamental Research Funds for the Central Universities (2019QNA3009, X.H.), the State Key Laboratory of Clean Energy Utilization (ZJUCEU2020007, X.H.). Calculations were performed on the high-performance computing system at the Department of Chemistry, Zhejiang University.

## Author contributions
Y.-X.J. and X.H. conceived the idea and supervised the whole project. B.Z., Z.-Y.C., J.-W.Z., R.-X.L. designed and carried out the experiments. H.W. carried out the DFT calculation. All authors discussed the results, contributed to writing and commented on the paper, and approved the final version of the paper for submission.

## Competing interests
The authors declare no competing interests.
