## [Peer Review File · Nature Communications]

REVIEWER COMMENTS

Reviewer #1 (Remarks to the Author):

The authors report palladium catalyzed intramolecular insertion of aryl halides on naphthalenes which is followed by coupling with tetraaryl borates. DFT calculations also supports the putative reaction pathway. In another example, alkyne insertion was incorporated before 5-exo-trig cyclization onto naphthalene.

Although chiral ligands are used (Table 1), no ee values are included in the manuscript which is somewhat unusual. I will reconsider the manuscript if the ligand control of absolute configuration is added to the main text.

The usefulness of the products from insertion of naphthalene seems to be a myth.

Reviewer #2 (Remarks to the Author):

As an efficient synthetic method converting planar arene molecules to alicyclic compounds, the dearomative functionalization reaction of arene has received considerable attention. Among, Heck-anionic capture sequence involving initial dearomative carbometallation step has been applied to the dearomative difunctionalization of heteroarenes, such as indoles and furans. The reaction of aromatic carbocycle has remained underexplored and challenging as its relatively higher resonance stabilization energy. In this communication, the authors report the first example of palladium-catalyzed dearomative difunctionalization of naphthalene, which was realized through a domino Heck-Suzuki coupling reaction. A few competitive side reactions, including intramolecular naphthalenyl C-H arylation, intermolecular Suzuki cross-coupling of Ar-Br with organoboron compounds, and dearomative 1,2-functionalization reaction, have been successfully inhibited based on the development of a new and unique phosphoramidite ligand as well as careful reaction condition optimization. Very good diastereoselectivities and 1,4-regioselectivities with good to excellent yields have been observed in the formation of a large number of dihydronaphthalene-based spirocyclic compounds. Moreover, the DFT calculations has been conducted to understand the reaction pathway and showed the controlling factors of chemo- and regioselectivities. Overall, this work represents an important and good step forward in the field of dearomatization reaction. This review is very willing to recommend its acceptance for publication in Nature Communication after the following minor revisions being considered.

1) The authors do not provide the information about the enantiocontrol of the reaction. What's the ee value of the standard product during condition optimization if the corresponding chiral ligands were used?

2) What's the result for the reaction using L4 as ligand and KF as base in toluene?

3) The molecular structure of 1w seems to be wrong as it is as same as 1j.

4) What's the influence of counter anion in Ar-Pd-X species to the reaction yield? The authors should screen chloro-, iodo-, or TfO- instead of Br- in the standard substrate.

Reviewer #3 (Remarks to the Author):

In this manuscript, Hong, Jia and coworkers describe a Pd-catalyzed dearomatic 1,4-difunctionalization of naphthalene derivatives with arylboronic acids and intramolecular aryl bromides. The reaction forms complex spirobicyclic products in good to excellent yields. The authors have also expanded the chemistry to the addition of alkynes followed by dearomatic arylation. The authors have also conducted DFT calculations to determine the reaction pathway. The manuscript is also well-written and easy to follow. This is a very good work and I recommend its publication but after some critical revisions as noted below:

1. The background discussion in the first paragraph is generally very good. However, the critical review on alkene dicarbofunctionalization cited as ref 13, which is the most relevant to the current work, should be moved up to the first paragraph along with refs 1-8.
2. The comparison of the current difunctionalization to diene is somewhat inaccurate. The authors should treat it as a reaction in a broader class of dicarbofunctionalization and draw a general alkene dicarbofunctionalization equation in Figure 1 with citation of the most relevant ref 13.
3. The reaction is conducted only with naphthalene derivatives. The authors should comment on what happens with benzene derivatives. It's likely that the reaction also works with higher member arenes such as anthracene, phenanthrene etc. The authors should include results or comment on what happens with benzene derivatives and the higher member arenes.
4. The reaction scope is also explored only with very tolerant functional groups such as CF₃, F, OMe, Me etc (except one ester). The authors should show or comment on what happens with naphthalenes bearing CN, COMe, CONR₂ etc.
5. What happens if the para-position is occupied? Does the reaction form 1,2-difunctionalized products? This would be very interesting in terms of substrate scope.

Reviewer #4 (Remarks to the Author):

Jia and Hong developed an efficient dearomative 1,4-diarylation of naphthalenes via Pd-catalyzed tandem Heck/Suzuki coupling reaction. Through ingenious design of experimental methods, a few competitive side reactions are overcome with good selectivity. The method further extended to three-component dearomative 1,4-vinylarylation of naphthalenes with alkynes and tetraarylborates. The origins of chemo- and regioselectivities have been elucidated by DFT calculations. It is the facile exergonic insertion of naphthalene double bond that disrupts the sequence of direct Suzuki coupling, leading to the tandem Heck/Suzuki coupling reaction. The steric repulsions between the aryl group of the forming C–C bond and the spiro-scaffold disfavors the intrinsic regioselectivity towards 1,2-difunctionalization, allowing the observed 1,4-difunctionalization. This manuscript is very interesting and can be accepted by Nature Communications after minor revision.

(1) The calculated results well explain the selectivity of the product, why M06-L function is selected for energy calculations.

(2) TS7 is slightly more favorable than that of TS11, and the stability of two transition states can be compared from the charge analysis or others ?

(3) The effect of aryl halides or pseudohalide to the reaction is suggested to be investigated. It's better to provide the results of the reactions of aryl chloride and aryl iodide, or ArOTf if possible under the optimal condition. Moreover, only two types of boron-coupling agents of PhB(OH)₂ and Ph₄BNa were tested during condition optimization, others such as (PhO)₃B or PhBpin should be further investigated.

(4) Ar₄BNa was used as coupling reagent in the three-component sequential reactions showed in Table 3. What's the results if commercially much available ArB(OH)₂ was used instead of Ar₄BNa. In this reaction, only symmetric alkynes were studied. Could the unsymmetric alkynes be used?

(5) What's the result if other classic nucleophilic organometallic reagents were used as trapping agents, such as alkylboron, organosilicon, or organotin reagents?

(6) If beta-naphthamide substrate was used instead of the present alpha-naphthamides, would the 1,2-diarylation reaction occur?

(7) Excellent selectivity of 1,4- vs 1,2- have been observed in the reaction. While if substituent

occupied C4-position of the naphthalene ring, could the 1,2-diarylation product be achieved in enhanced yield?

Point-by-point Response to Referees

Reviewer #1 (Remarks to the Author):

The authors report palladium catalyzed intramolecular insertion of aryl halides on naphthalenes which is followed by coupling with tetraaryl borates. DFT calculations also support the putative reaction pathway. In another example, alkyne insertion was incorporated before 5-exo-trig cyclization onto naphthalene.

Although chiral ligands are used (Table 1), no ee values are included in the manuscript which is somewhat unusual. I will reconsider the manuscript if the ligand control of absolute configuration is added to the main text.

The usefulness of the products from insertion of naphthalene seems to be a myth.

Response:

We thank this reviewer very much for the question about the enantiocontrol of this reaction. We initially used chiral ligands to study the enantioselective reaction. Unfortunately, a number of chiral ligands tested lead to lower enantioselectivities (Ee values for the reactions using chiral L1-L5 ligands are now provided in the main text; Table 1, entry 4 (*S*)-L1 16% ee; entry 5 (*R*)-L2, 8% ee; entry 6 (*S*)-L3, 12% ee; entry 7 (*S*)-L4, 11% ee; entry 20 (*R*)-L5, 5% ee). Therefore, in this communication we mainly focused on the diastereoselective dearomative Heck-Suzuki reaction by overcoming a few very competitive side reactions (the intramolecular naphthalenyl C-H arylation, the intermolecular Suzuki cross-coupling, the dearomative 1,2-difunctionalization, and the dearomative reductive-Heck reaction). Key to the present study relies on the finding of a unique spiro-phosphoramidite ligand. Three types of 1,4-dihydronaphthalene-based spirocyclic compounds are achieved in excellent regio- and diastereoselectivities with moderate to excellent yields, showing a broad substrate scope. Spirooxindole represents an important structural subunit

of alkaloids and other bioactive molecules. Spirooxindole molecules achieved by simple transformations of product 3a are analogues to the core structure of a patented bioactive molecule for treating pain.

Reviewer #2 (Remarks to the Author):

As an efficient synthetic method converting planar arene molecules to alicyclic compounds, the dearomative functionalization reaction of arene has received considerable attention. Among, Heck-anionic capture sequence involving initial dearomative carbometallation step has been applied to the dearomative difunctionalization of heteroarenes, such as indoles and furans. The reaction of aromatic carbocycle has remained underexplored and challenging as its relatively higher resonance stabilization energy. In this communication, the authors report the first example of palladium-catalyzed dearomative difunctionalization of naphthalene, which was realized through a domino Heck-Suzuki coupling reaction. A few competitive side reactions, including intramolecular naphthalenyl C-H arylation, intermolecular Suzuki cross-coupling of Ar-Br with organoboron compounds, and dearomative 1,2-functionalization reaction, have been successfully inhibited based on the development of a new and unique phosphoramidite ligand as well as careful reaction condition optimization. Very good diastereoselectivities and 1,4-regioselectivities with good to excellent yields have been observed in the formation of a large number of dihydronaphthalene-based spirocyclic compounds. Moreover, the DFT calculations has been conducted to understand the reaction pathway and showed the controlling factors of chemo- and regioselectivities. Overall, this work represents an important and good step forward in the field of dearomatization reaction. This review is very willing to recommend its acceptance for publication in Nature Communication after the following minor revisions being considered.

1) The authors do not provide the information about the enantiocontrol of the reaction. What's the ee value of the standard product during condition optimization if the corresponding chiral ligands were used?

Response:

We really thank this reviewers' question about the ee value of the standard product in the condition optimization. We initially used chiral ligands to study the enantioselective reaction, while lower enantioselectivities were observed for a number of chiral ligands. In the revised manuscript, ee values for the reactions using chiral L1-L5 ligands are now provided in the main text.

2) What's the result for the reaction using L4 as ligand and KF as base in toluene?

Response:

In the presence of Pd(dba)₂ (5 mol%), L4 (10 mol%), and KF (2.5 equiv.), the reaction of 1a with PhB(OH)₂ (2.0 equiv.) in toluene at 120 °C for 24 h led to target product 3a in 71% NMR yield along with direct Suzuki coupling product 4 in 21% yield and 1,2-dearomative diarylation product 6 in 3% yield (Determined by 1H NMR spectroscopy using MeNO₂ as an internal standard). These results are now added in entry 13 of Table 1 in the revised main text.

3) The molecular structure of 1w seems to be wrong as it is as same as 1j.

Response:

We thank this review for the correction of the structure of 1w, which has been changed in the revised main text.

4) What's the influence of counter anion in Ar-Pd-X species to the reaction yield? The authors should screen chloro-, iodo-, or TfO- instead of Br- in the standard substrate.

Response:

The chloro-, iodo-, and TfO-substrates have been tested in the reactions with Ph₄BNa under the optimal conditions. The desired product was obtained in 89% NMR yield for the iodo-substrate. The chloro-substrate was relatively inert to give the product in 27% NMR yield, while no reaction took place for TfO-substrate. These results have been added in entries 22 and 23 in table 1.

Reviewer #3 (Remarks to the Author):

In this manuscript, Hong, Jia and coworkers describe a Pd-catalyzed dearomatic 1,4-difunctionalization of naphthalene derivatives with arylboronic acids and intramolecular aryl bromides. The reaction forms complex spirobicyclic products in good to excellent yields. The authors have also expanded the chemistry to the addition of alkynes followed by dearomatic arylation. The authors have also conducted DFT calculations to determine the reaction pathway. The manuscript is also well-written and easy to follow. This is a very good work and I recommend its publication but after some critical revisions as noted below:

1. The background discussion in the first paragraph is generally very good. However, the critical review on alkene dicarbofunctionalization cited as ref 13, which is the most relevant to the current work, should be moved up to the first paragraph along with refs 1-8.

Response:

We thank this review for this suggestion. Ref. 13 is now changed to ref. 9 in the first sentence.

2. The comparison of the current difunctionalization to diene is somewhat inaccurate. The authors should treat it as a reaction in a broader class of dicarbofunctionalization and draw a general alkene dicarbofunctionalization equation in Figure 1 with citation of the most relevant ref 13.

Response:

We thank this review very much for the suggestion. A general equation about transition-metal-catalyzed alkene difunctionalization is now provided, which includes the simple alkenes and dienes. The most relevant references 2 and 9-16 are highlighted.

3. The reaction is conducted only with naphthalene derivatives. The authors should comment on what happens with benzene derivatives. It's likely that the reaction also works with higher member arenes such as anthracene, phenanthrene etc. The authors should include results or comment on what happens with benzene derivatives and the higher member arenes.

Response:

We thank this review very much for the comments. We have tried the reactions of benzene or anthracene substrates. As shown below, the dearomative 1,4-diarylation reaction of benzene was tested under the standard condition. The desired 1,4-diarylation product could be detected but with a poor yield and the major product is derived from direct Suzuki-coupling reaction (95% isolated yield for the mixture, 1:10 by ¹H NMR). Comment of this result has been in the main text.

N-([1,1'-biphenyl]-2-yl)-*N*-methylbenzamide

Purified by chromatography on silica gel, eluting with ethyl acetate/petroleum ether 1:15 (v/v); white solid, Mp = 108-110 °C; ¹H NMR (600 MHz, CDCl₃): δ 7.51-7.28

(m, 7.00H), 7.25-7.17 (m, 1.86H), 7.04-7.01 (m, 3.47H), 6.92 (d, $J = 7.8$ Hz, 1.72H), 3.44 (s, 2.66H), 2.91 (s, 0.34H). ^{13}C NMR (150 MHz, CDCl_3): δ 169.9, 142.3, 139.1, 138.6, 135.4, 131.3, 129.4, 128.6, 128.5, 128.46, 128.44, 128.3, 127.7, 127.4, 127.1, 39.0. HRMS m/z (ESI $^+$): Calculated for $\text{C}_{20}\text{H}_{18}\text{NO}$ ($[\text{M}+\text{H}]^+$): 288.1383, found 288.1397.

^1H NMR of the mixture

The reaction of anthracene-substrate **1x** was also tested, which led to the desired 1,4-diarylation product **3x** in 88% isolated yield under the standard conditions. This result has been added in Figure 3.

4. The reaction scope is also explored only with very tolerant functional groups such as CF_3 , F, OMe, Me etc (except one ester). The authors should show or comment on what happens with naphthalenes bearing CN, COMe, CONR_2 etc.

Response:

We thank this review very much for the comments. The reactions of substrates bearing COMe (**1l**) and CONEt_2 (**1s**) were tested under the standard conditions,

which smoothly afforded the corresponding 1,4-diarylation products 3l and 3s in 64% and 90% yield, respectively. These results have been added in Figure 3.

5. What happens if the para-position is occupied? Does the reaction form 1,2-difunctionalized products? This would be very interesting in terms of substrate scope.

Response:

We thank very much for this review's suggestion. We tried the reaction of the substrate 8 bearing a *para*-MeO group on the naphthalene ring. As observed, the 1,4-diarylation product 9 having two tetrasubstituted stereocenters was isolated in 51% yield as the major product along with 1,2-product in 22% yield. This result has been added in Figure 3.

Reviewer #4 (Remarks to the Author):

Jia and Hong developed an efficient dearomative 1,4-diarylation of naphthalenes via Pd-catalyzed tandem Heck/Suzuki coupling reaction. Through ingenious design of experimental methods, a few competitive side reactions are overcome with good selectivity. The method further extended to three-component dearomative 1,4-vinylarylation of naphthalenes with alkynes and tetraarylborates. The origins of chemo- and regioselectivities have been elucidated by DFT calculations. It is the facile exergonic insertion of naphthalene double bond that disrupts the sequence of direct Suzuki coupling, leading to the tandem Heck/Suzuki coupling reaction. The steric repulsions between the aryl group of the forming C–C bond and the

spiro-scaffold disfavors the intrinsic regioselectivity towards 1,2-difunctionalization, allowing the observed 1,4-difunctionalization. This manuscript is very interesting and can be accepted by Nature Communications after minor revision.

(1) The calculated results well explain the selectivity of the product, why M06-L function is selected for energy calculations.

Response:

Thanks the referee for the supportive comments. M06-L is a local density functional, which has good performance for both main-group chemistry and transition metal chemistry [J. Chem. Phys. 2006, 125, 194101]. A number of previous computational studies [J. Am. Chem. Soc., 2016, 138, 2712; J. Am. Chem. Soc., 2017, 139, 5194; J. Am. Chem. Soc., 2017, 139, 3546.] also supported that this functional is appropriate to describe palladium-catalyzed reactions. To further confirm the reliability of M06-L results, we also verified the free energy profile using wB97XD functional (Figure S5). The same chemoselectivity was identified.

Figure 1. DFT-computed free energy profiles of the most favorable pathway for the formation of **3a** and by-product **4** (L = L5), with single point energy calculated using wB97XD functional.

Figure 2. DFT-computed free energy profiles of the most favorable pathway for the formation of **3a** and by-product **4** ($L = L5$), with single point energy calculated using M06-L functional.

(2) TS7 is slightly more favorable than that of TS11, and the stability of two transition states can be compared from the charge analysis or others?

Response:

TS7 is more stable than TS11 due to the intrinsic stability of the spiro scaffold. From int3, the alkene insertion via TS4 is exergonic by 11.2 kcal/mol, leading to the more stable spiro intermediate int5. This stability of spiro scaffold is carried in the subsequent pre-transmetalation intermediates (int6 vs. int10) as well as the transmetalation transition states (TS7 vs. TS11). Therefore, the facile exergonic insertion of pendant naphthalene double bond via TS4 interrupts the sequence of Suzuki cross-coupling and drives the reaction towards the tandem Heck/Suzuki coupling reaction.

(3) The effect of aryl halides or pseudohalide to the reaction is suggested to be investigated. It's better to provide the results of the reactions of aryl chloride and aryl iodide, or ArOTf if possible under the optimal condition. Moreover, only two types of boron-coupling agents of PhB(OH)₂ and Ph₄BNa were tested during condition optimization, others such as (PhO)₃B or PhBpin should be further investigated.

Response:

The chloro-, iodo-, and TfO-substrates have been tested in the reactions with Ph₄BNa under the optimal conditions. The desired product was obtained in 89% NMR yield for the iodo-substrate. The chloro-substrate was relatively inert to give the product in 27% NMR yield, while no reaction took place for TfO-substrate. The results have been added in entries 22 and 23 in table 1.

Moreover, PhBPin and (PhBO)₃ were also tested as boron-coupling agents to react with substrate 1a under the standard conditions. As shown below no reaction was observed for PhBPin and inferior results were observed for (PhBO)₃.

(4) Ar₄BNa was used as coupling reagent in the three-component sequential reactions showed in Table 3. What's the results if commercially much available ArB(OH)₂ was used instead of Ar₄BNa. In this reaction, only symmetric alkynes were studied. Could the unsymmetric alkynes be used?

Response:

As shown below, when PhB(OH)₂ was used instead of Ph₄BNa in the reaction showed in Table 3, the desired product was only obtained in 32% yield. This implies PhB(OH)₂ is much less active than Ph₄BNa. When an unsymmetric alkyne was used, the desired product was isolated in 53% yield as a 1:1 mixture.

The 1:1 mixture of 2-Pentyl-4'-phenyl-3-propyl-4'*H*-spiro[indene-1,1'-naphthalene] and 3-Pentyl-4'-phenyl-2-propyl-4'*H*-spiro[indene-1,1'-naphthalene]

Purified by chromatography on silica gel, eluting with petroleum ether; pale yellow oil, 44.4 mg, 53% yield; $^1\text{H NMR}$ (600 MHz, CDCl_3): δ 7.35-7.31 (m, 4H), 7.30-7.26

(m, 6H), 7.26-7.21 (m, 4H), 7.07-6.98 (m, 8H), 6.91-6.87 (m, 2H), 6.50-6.47 (m, 2H), 6.09-6.06 (m, 2H), 5.25-5.22 (m, 2H), 4.90 (s, 2H), 2.63-2.56 (m, 4H), 2.39-2.31 (m, 2H), 2.20-2.13 (m, 2H), 1.76-1.66 (m, 4H), 1.44-1.39 (m, 6H), 1.31-1.19 (m, 6H), 1.06 (t, $J = 7.2$ Hz, 3H), 0.94 (t, $J = 6.6$ Hz, 3H), 0.86 (t, $J = 7.8$ Hz, 3H), 0.81 (t, $J = 7.2$ Hz, 3H). ^{13}C NMR (150 MHz, CDCl_3): δ 154.29, 154.26, 151.7, 151.2, 146.2, 144.8, 138.7, 138.3, 137.24, 137.22, 135.27, 135.21, 130.0, 129.39, 129.37, 128.84, 128.80, 128.6, 128.5, 127.89, 127.82, 127.1, 126.75, 126.73, 126.48, 126.43, 126.3, 125.04, 125.00, 123.73, 123.72, 118.7, 59.27, 59.21, 45.37, 45.36, 32.4, 32.2, 29.6, 29.2, 28.7, 27.7, 27.0, 25.7, 23.2, 22.6, 22.3, 22.2, 14.6, 14.5, 14.1, 14.0. HRMS m/z (EI+): Calculated for $\text{C}_{32}\text{H}_{34}(\text{M}^+)$: 418.2661, found 418.2681.

(5) What's the result if other classic nucleophilic organometallic reagents were used as trapping agents, such as alkylboron, organosilicon, or organotin reagents?

Response:

Followed this suggestion, we have tested other nucleophilic organometallic reagents. No reaction took place for $\text{PhSi}(\text{OMe})_3$, $\text{CyB}(\text{OH})_2$, $\text{MeB}(\text{OH})_2$. When PhSn^nBu_3 was used in the reaction with **1a**, product **3a** was observed in a lower NMR yield (24%).

1a: **3a**: **5** = 65%: 24%: 3% (Determined by ^1H NMR)

(6) If beta-naphthamide substrate was used instead of the present alpha-naphthamides, would the 1,2-diarylation reaction occur?

Response:

We have tried the reaction of beta-naphthamide substrate, while only the direct Suzuki-coupling product was isolated in 87% yield.

N-([1,1'-biphenyl]-2-yl)-*N*-methyl-2-naphthamide

Purified by chromatography on silica gel, eluting with ethyl acetate/petroleum ether 1:10 (*v/v*); pale yellow oil, 58.7 mg, 87% yield; ^1H NMR (600 MHz, CDCl_3): δ 7.72 (d, $J = 8.4$ Hz, 1H), 7.58-7.33 (m, 9H), 7.31-7.26 (m, 2H), 7.19 (d, $J = 7.2$ Hz, 1H), 7.00 (d, $J = 8.4$ Hz, 1H), 6.93 (d, $J = 7.2$ Hz, 2H), 3.52 (s, 3H). ^{13}C NMR (150 MHz, CDCl_3): δ 169.9, 142.3, 139.2, 138.6, 133.4, 132.9, 132.1, 131.3, 129.4, 128.7, 128.5, 128.5, 128.3, 127.7, 127.43, 127.41, 126.9, 126.5, 126.0, 125.3, 39.1. HRMS m/z (ESI $^+$): Calculated for $\text{C}_{24}\text{H}_{20}\text{NO}$ ($[\text{M}+\text{H}]^+$): 338.1539, found 338.1540.

(7) Excellent selectivity of 1,4- vs 1,2- have been observed in the reaction. While if substituent occupied C4-position of the naphthalene ring, could the 1,2-diarylation product be achieved in enhanced yield?

Response:

We thank very much for this review's suggestion. We tried the reaction of the substrate **8** bearing a *para*-MeO group on the naphthalene ring. As observed, the 1,4-diarylation product **9** bearing two tetrasubstituted stereocenters was isolated in 51% yield as the major product along with 1,2-product in 22% yield. This result has been added in Figure 3.

REVIEWERS' COMMENTS:

Reviewer #2 (Remarks to the Author):

The authors address the questions raised by me clearly. As a result, I recommend its publication in Nature Communications without further revision.

Reviewer #3 (Remarks to the Author):

The authors have addressed all prior concerns and with the revision incorporating all the changes, the quality of the manuscript has significantly increased. I recommend its publication in its current form.

Reviewer #4 (Remarks to the Author):

This manuscript has been greatly improved by Jia and Hong, I think now it can be accepted by Nature Communications.

RESPONSE to REVIEWERS' COMMENTS

Reviewer #2 (Remarks to the Author):

The authors address the questions raised by me clearly. As a result, I recommend its publication in Nature Communications without further revision.

Response:

We thank very much for this reviewer's valuable comments, which help us to improve the quality of this manuscript.

Reviewer #3 (Remarks to the Author):

The authors have addressed all prior concerns and with the revision incorporating all the changes, the quality of the manuscript has significantly increased. I recommend its publication in its current form.

Response:

We thank very much for this reviewer's valuable comments, which help us to improve the quality of this manuscript.

Reviewer #4 (Remarks to the Author):

This manuscript has been greatly improved by Jia and Hong, I think now it can be accepted by Nature Communications.

Response:

We thank very much for this reviewer's valuable comments, which help us to improve the quality of this manuscript.